# Cadmium and volumetric mammographic density: A cross-sectional study in Polish women

**Beata Pepłońska**[1]*, **Beata Janasik**[2], **Valerie McCormack**[3], **Agnieszka Bukowska-Damska**[1], **Paweł Kałużny**[1]

**1** Department of Environmental Epidemiology, Nofer Institute of Occupational Medicine, Lodz, Poland,
**2** Department of Biological and Environmental Monitoring, Nofer Institute of Occupational Medicine, Lodz, Poland, **3** Section of Environment and Radiation, International Agency for research on Cancer, Lyon, France

* Beata.Peplonska@imp.lodz.pl

## Abstract

### Introduction

Cadmium (Cd) is a heavy metal, which is widespread in the environment and has been hypothesized to be a metalloestrogen and a breast cancer risk factor. Mammographic density (MD) reflects the composition of the breast and was proposed to be used as a surrogate marker for breast cancer. The aim of our study was to investigate association between cadmium concentration in urine and mammographic density.

### Methods

A cross-sectional study included 517 women aged 40–60 years who underwent screening mammography in Łódź, Poland. Data were collected through personal interviews and anthropometric measurements. Spot morning urine samples were obtained. The examination of the breasts included both craniocaudal and mediolateral oblique views. Raw data ("for processing") generated by the digital mammography system were analysed using Volpara Imaging Software, The volumetric breast density(%) and fibrograndular tissue volume (cm³) were determined. Cadmium concentration in urine was analysed using the standard ICP-MS method.

### Results

After adjusting for key confounders including age, BMI, family breast cancer, mammographic device, season of the year of mammography, and age at menarche, an inverse association of Cd and volumetric breast density was found, which was attenuated after further adjustment for smoking. Associations of Cd with dense volume were null.

### Conclusions

These findings suggest that Cd is not positively associated with breast density, a strong marker of breast cancer risk, when examined in a cross-sectional fashion.

**Data Availability Statement:** The datasets generated and/or analysed during the current study

are available in the ecnis openrepository: http://hdl.handle.net/10146/618315.

**Funding:** BP received the grant no UMO-2015/17/B/NZ7/02928 of the National Science Centre in Poland (www.ncn.gov.pl). The project was also financed by the statute activity of Nofer Institute of Occupational Medicine in Lodz No. IMP 10.16 and IMP 10.34 The funders had no role in study design, data collection and analysis, decision to publish, or preparation of the manuscript.

**Competing interests:** The authors declare they have no actual or potential competing financial interests. Where authors are identified as personnel of the International Agency for Research on Cancer/World Health Organization, the authors alone are responsible for the views expressed in this article and they do not necessarily represent the decisions, policy or views of the International Agency for Research on Cancer / World Health Organization.

## Introduction

Cadmium (Cd) is a heavy metal which is widespread in the environment. Occupational exposure to cadmium occurs in many occupational settings, such as pigment and battery production, galvanization and recycling of electric tools. Environmental contamination with Cd originates from industrial sources and the use of high-cadmium fertilizers in agriculture. In the general population, food (especially vegetables and offal) is the major source of Cd exposure [1].

In humans, the body burden of cadmium accumulates with increasing age and differs across world regions. Relatively high levels were found in Japan, while lower levels in Europe and the US [2, 3]. Environmental exposures to Cd were evaluated in the course of the European multicentre human biomonitoring project COPHES/DEMOCOPHES in mother-child pairs. Cd exposures were measured among 1632 women from 16 European countries [2], including Poland. Overall, creatinine (Cr)-adjusted Cd medians in spot morning urine samples amounted to 0.24 μg/gCr in smokers and 0.18 μg/gCr in nonsmokers. However, in Polish women, the Cd concentrations were the highest (~0.42 μg/gCr and 0.36 μg/gCr respectively), that is double the European medians in both smokers and nonsmokers. The relatively high Cd concentrations found among Poles were explained by the high contamination of farmland from fertilizers with a high Cd content.

Cd accumulates in the body and has a long half-life of about 10–30 years. Approximately 50% of the total body burden of cadmium is accumulated in the kidneys. Urinary cadmium concentrations are thought to reflect exposure over a period of 20–30 years [1].

Cadmium has been classified by the International Agency for Research on Cancer as carcinogenic to humans (Group 1) (International Agency for Research on Cancer, 2012) and has been linked to renal, lung and prostate cancer. Over the past decade, there has been an increasing interest in Cd as a potential risk factor for breast cancer. Of relevance, relatively high Cd concentrations have been observed in women's breast tissue, 20–30 μg/g [4]. The proposed mechanism includes Cd as a metalloestrogen [4], but this pathway requires further investigation. A systematic review of epidemiologic studies investigating association between urinary Cd concentrations and breast cancer risk showed a positive association, with a combined odds ratio of 1.66 (95% CI: 1.23–2.25) per 0.5 μg/g Cr increase in Cd concentration. This finding referred to case-control and cross-sectional studies. However, the findings from the cohort studies were scarce and rather inconsistent [5]. Prospective cohort studies did not confirm positive association between cadmium and breast cancer risk [6, 7]. Moreover, an inverse association between cadmium levels in stored erythrocytes and breast cancer risk was found when three prospective cohorts were analyzed [7]

Mammographic density (MD) reflects the fibroglandular composition of the breast. It is positively associated with collagen, epithelial and non-epithelial cells, and negatively associated with fat [8]. MD has been found to be a strong and independent risk factor for breast cancer, with a 4.6–fold increased breast cancer risk observed in women with extensive mammographic density (>75%), when compared to women with a small proportion of dense areas in the breast (<5%) [9]. High density was suggested to share some common risk factors with breast cancer [10]. MD is modifiable, it changes over time as well as during postmenopausal hormone therapy [11] Strong associations between MD and some factors such as postmenopausal hormone use and tamoxifen therapy [12], inversed relationships with age, postmenopausal status, and higher BMI [9] are well established. Further potential influences including association between MD and for example alcohol consumption [13], smoking, physical activity [14], night shift work [15] and other occupational exposures [16] are being investigated worldwide.

Whilst it has been postulated that MD may serve as a powerful intermediate biomarker of breast cancer risk, with which we can investigate the cumulative impact of environmental exposures such as endocrine disrupting chemicals (EDC) [17], we have identified only two cross-sectional studies examining the association between MD and Cd. One of these, conducted on 190 premenopausal women, reported a significant positive association between urinary Cd and MD, with MD measured using the area-based percent MD (via Cumulus) as well as BI-RADS [18]. Interestingly, in a larger study of 725 women, the findings were null [19]. The latter study however, was based on the BI-RADS 4-category classification which may be too crude to detect small effects. Thus, to further explore the potential role of Cd in the modulation of MD, we undertook a cross-sectional study of MD in Polish women, in whom exposure contrasts are expected to be higher than in previous studies. The study additionally benefits from a volumetric breast density (VBD) assessment method, which is a fully automatic, objective and continuous measure using the *Volpara* software.

## Materials and methods

### Study design and population

We conducted a cross-sectional study of MD in relation to cadmium in a population of women from the city of Łódź, central Poland. Women were recruited into the study at two mammographic screening centres at the time they were presenting voluntarily for screening mammography. Women were eligible for study inclusion if they were 40–60 years of age, residents of Łódź area, had no previous diagnosis of breast cancer or previous breast augmentation surgery/implants and at the time of enrollment declared they were not on hormone replacement therapy (HRT). This sampling frame is population-based as the national mammography screening is funded from the government to women aged 50–69 every two years. The programs for women aged 40–49 are also carried out, but on a minor scale and on irregular basis. Women were enrolled in the study during 2013–2018 when 600 women, initially classified as eligible, provided consent to participate. Out of these women, 83 were excluded: n = 43 later refused, n = 2 did not provide urine sample and n = 6 reported using HRT during the interview. For n = 32 women, mammographic images had not been recorded in the raw, "for processing", format required for volumetric density calculations. Finally, the data of 517 women were included in the analysis.

Initial sample size considerations for this study were based on the previous study by Adams et al. [18]. We assumed the difference in cadmium means of 26% (inferred from [18]) between the group with the highest breast density as compared to those with lower density (SD = 0.26), two-sided t-test, and significance level 0.05 and power 0.8, which yielded 132 per group (tertiles: low, mid and high MD). Taking account of the potential differences between the populations (with respect to age, MD, and Cd levels) we enlarged the sample size by 25%. Eventually, the total study population included 500 women.

Personal interviews were carried out at women's homes (on average within 1.5 month since mammography) by trained interviewers to elucidate data on demographics, menstruation and menopause, reproductive history, contraceptive medications usage history, menopausal hormone therapy, alcohol consumption, and tobacco smoking. Women were provided with polyethylene, 50 ml volume, urine containers, which were washed in 20% nitric acid (24h) and then rinsed with ultrapure water (Milli-Q Integral 3, Merck, Poland) to avoid contamination. Participants collected a spot morning urine sample at their homes and brought it back to the centre, with a median of 24 days after mammography.

Anthropometric measurements i.e. body weight and height, hip and waist circumferences were carried out by trained nurses, on average within one month after mammography. Body

mass index (BMI) (body weight divided by squared height, in kg/m$^2$), and waist to hip ratio (WHR) (waist circumference at umbilical in cm divided by hip circumference) were calculated.

### Ethics statement

The study was approved by the Bioethics Committee at Nofer Institute of Occupational Medicine (approval no. 2/2012 of 13th March, 2012). A signed informed consent was obtained from each study participant.

### Mammography and mammographic density assessment

Digital mammography was performed in two mammographic centers, according to standard procedure, with Mammomat Novation DR, Mammomat Fusion ((Siemens Healthcare GmbH, Germany) in one center, and Lorad Selenia, Selenia Dimensions (Hologic Inc. USA) in the other. The examination of the breasts included both craniocaudal and mediolateral oblique views. Raw data ("for processing") generated by the digital mammography system were analysed using Volpara Imaging Software (Volpara Health Technologies Ltd., Wellington, New Zealand), algorithm version 1.5.5.1, at the Department of Environmental Epidemiology NIOM. Volpara applies a physics-based model, and its principles were described by Highnam et al. [20] as an extension of the method proposed by van Engeland et al. [21]. Briefly, the algorithm determines the x-ray attenuation between the image detector and the x-ray source according to the image pixel signal. A pixel intensity corresponding to purely adipose tissue is used as a reference to which all other pixels are compared to calculate the thickness of the fibroglandular tissue that must have been present to contribute to a relatively greater x-ray attenuation than at the fatty reference point. The volumes of the adipose and fibroglandular tissue are summed across the entire breast. Volumetric breast density (VBD) is calculated as the ratio of the fibroglandular tissue volume to the total breast volume, and is expressed as percentage. In this, for each women (combining her 4 views) this quantitative VBD value is mapped to one of four Volpara Density Grades (VDG) based on thresholds (VDG a $<$ 3.5% VBD; VDG b $\geq$ 3.5% and $<$ 7.5% VBD; VDG c $\geq$ 7.5% and $<$ 15.5% VBD; VDG d, $\geq$ 15.5% VBD) such that the VDG categories correlate with the density categories (a, b, c, and d) listed in the American College of Radiology Breast Imaging-Reporting and Data System 5th edition density categories [22, 23].

### Urinary cadmium analysis

An ELAN$^®$ DRC-e ICP-MS (PerkinElmer SCIEX, USA), equipped with a Mainhard quartz nebulizer, quartz cyclonic spray chamber and platinum sampler and skimmer cones, was used for cadmium (Cd) analysis in urine. Cadmium ($^{114}$Cd) was analysed at the Department of Biological and Environmental Monitoring, NIOM, using the standard ICP-MS method and the Dynamic Reaction Cell (DRC-ICP-MS) which eliminates molybdenum oxide interferences. The DRC parameters were 1.0 mL/min methane (Linde Gas, Poland) flow rate and 0.85 RPq. Prior to analysis, the samples were centrifuged, and supernatants (0.2 ml) were diluted with 1.8 mL of diluent (1% nitric acid, 70%, ULTREX™ II Reagent, J.T.Baker™, Witko, Poland). External calibration ranges were 0.1–10 µg/L for cadmium (Multi-Element Calibration Standard, Perkin Elmer Pure Plus, Poland). Clinchek$^®$ urine (Recipe, Germany) was analysed every 10 samples as an internal quality control check. The performing laboratory participates in the external quality program for cadmium in urine analysis, which is coordinated by the Institute of Occupational, Social and Environmental Medicine of the University of Erlangen, Nuremberg, Germany (G-EQUAS).

### Creatinine determination

Creatinine content was determined using colorimetric Jaffe method [24]. The analysis was carried out at 520 nm on Cary 60 UV-Vis Agilent Technologies spectrometer (MS Spektrum, Poland).

### Statistical analysis

Arithmetic means (for continuous variables) and frequencies (for categorical variables) were calculated in order to characterize the study population. The means and standard deviations for the estimates of fibroglandular tissue volume ($cm^3$), breast volume (cm3) and volumetric breast density (%) for the left and right breast, and their average, were determined. To examine whether Cd influences the measures of MD, we fitted linear (normal-error) regression models of MD. Both MD metrics and creatinine-normalized urinary cadmium concentration measurements were right skewed; thus, in such models, their values were transformed using natural logarithms. Cr-adjusted Cd concentrations were also fitted as categorical variables using quartiles.

Based on literature review, the following variables were considered as the potential confounders of MD-Cd associations: age at mammography (continuous), BMI (continuous), smoking (never, ex-, current smoker), menopausal status (pre-, postmenopausal), age at menarche ($\leq$12,13–14, $\geq$15 years), previous use of hormonal contraceptives (ever, never), parity (ever, never), number of pregnancies (continuous), breastfeeding (ever, never) and family history of breast cancer (yes, no). Women were classified as premenopausal if they reported having the last menstrual bleeding within the last 365 days, otherwise they were classified as postmenopausal. Additionally, the variables capturing possible variability due to the time and technique of mammographic data collection: calendar season of mammographic examination (January-March, April-June, July-September, October-December), mammographic center (1, 2), mammographic X-ray system (Siemens, Hologic) and mammographic device (apparatus) (Mammomat Novation DR, Mammomat Fusion, and Lorad Selenia, Selenia Dimensions) were analyzed.

Variables that had a significance level of p<0.15 in univariate linear regression models, with VBD as the outcome variable, were then examined in the multivariate models. These included age, BMI, family history of breast cancer, mammographic centre, device, calendar season, age at menarche, menopausal status, smoking. The stepwise variable selection with Akaike information criterion (AIC) was applied, with age, BMI, family breast cancer, mammographic device, season of the year for mammography, and age at menarche retained in the final model. The smoking status variable was reinserted in one variant of the model. Additionally, we ran sensitivity analysis adjusted for urinary creatinine instead of dividing cadmium concentrations by creatinine. Stratified analyses were run, by the smoking status, menopausal status, family history of breast cancer, parity, and three age groups:$\leq$50; >50-$\leq$55, >55. The likelihood ratio test was applied to determine the statistical significance of effect modification.

The R software (R Core Team, 2018) version 3.5.1 was used for statistical analyses.

## Results

The mean age of participants at the time of mammography was 54.6 (SD3.8) years and the mean BMI was 27.2 (SD 4.6) kg/m$^2$ (Table 1). The majority of women (77.6%) were postmenopausal, had menarche at age 13–14 (44.3%), were ever pregnant (83.6%), and among the parous women, 59.2% had ever breastfed. Approximately 27% of the subjects ever used hormonal contraceptives. As many as 10.3% of women reported to have family history of breast cancer. Most of the screening mammographies were performed during the last calendar quarter of the

**Table 1. The selected characteristics of the study population.**

| Characteristics | mean (SD) | N(%) 517 (100) |
|---|---|---|
| Mean age at mammography (years) | 54.6 (3.8) | |
| BMI (kg/m2) | 27.2 (4.6) | |
| Menopausal status | | |
| Pre- | | 116 (22.4) |
| Post- | | 401 (77.6) |
| Age at menarche | | |
| <12 | | 133 (25.7) |
| 13–14 | | 229 (44.3) |
| ≥15 | | 95 (18.4) |
| missing | | 60 (11.6) |
| Parity | | |
| Ever | | 432 (83.6) |
| Never | | 32 (6.2) |
| missing | | 53 (10.2) |
| Breastfeeding | | |
| Ever | | 306 (59.2) |
| Never | | 158 (30.6) |
| missing | | 53 (10.2) |
| Family history of breast cancer | | |
| Yes | | 53 (10.25) |
| No | | 411 (79.5) |
| missing | | 53 (10.25) |
| Smoking | | |
| Current | | 109 (21.1) |
| Past | | 137 (26.5) |
| Never smoker | | 226 (43.7) |
| missing | | 45 (8.7) |
| Calendar year season when mammography was performed | | |
| January-March | | 58 (11.2) |
| April-June | | 91 (17.6) |
| July–September | | 111 (21.5) |
| October–December | | 257 (49.7) |
| Hormonal contraceptives use | | |
| Ever | | 138 (26.7) |
| Never | | 326 (63.1) |
| missing | | 53 (10.2) |
| Mammographic centre | | |
| 1 | | 286 (55.3) |
| 2 | | 231 (44.7) |
| x-ray system | | |
| Siemens Fusion | | 122 (23.6) |
| Siemens Novation | | 109 (21.1) |
| Hologic | | 286 (55.3) |
| Mammographic device | | |
| Mammomat NovationDR | | 109 (21.1) |
| Mammomat Fusion | | 122 (23.6) |
| Lorad Selenia | | 138 (26.7) |

(*Continued*)

**Table 1.** (Continued)

| Characteristics | mean (SD) | N(%) 517 (100) |
|---|---|---|
| Selenia Dimensions | | 148 (28.6) |
| Fibroglandular tissue volume (cm3) | | |
| Left | 59.7(31.2) | |
| Right | 61.6(33.3) | |
| Breast volume (cm3) | | |
| Left | 920 (467) | |
| Right | 918 (485) | |
| Percent volumetric mammographic density | | |
| Left | 7.6 (4.5) | |
| Right | 7.9 (4.6) | |
| Average | 7.8 (4.5) | |
| Volpara Density Grade (VDG) | | |
| a (< 3.5%) | | 35 (6.8) |
| b (≥3.5% and < 7.5%) | | 254 (49.1) |
| c (≥(7.5% and < 15.5%) | | 180 (34.8) |
| d (≥ 15.5%) | | 48 (9.3) |
| Cadmium concentration in urine (µg/l) | 0.56(0.43) | |
| Cadmium concentration in urine (creatinine- adjusted) (µg/gCr) | 0.65(0.42) | |

year (~50%), while the least of them during January-March(~11%). The mean volume of the fibroglandular tissue and the total breast volume were found to be similar in the left and right breast, i.e. 59.7(SD 31.2) cm$^3$ and 61.5(SD 33.3) cm$^3$, and 920 (SD 467) cm$^3$ and 918 (SD 485) cm$^3$, respectively. The average VBD was 7.8% (SD 4.5), and 9.3% of women had the highest Volpara density grade (d). The mean cadmium urine concentration was 0.56 (SD 0.43) µg/l and creatinine-adjusted 0.65 (SD 0.42) (µg/gCr). Urinary cadmium concentration was positively associated with age ($p<0.01$), postmenopausal status ($p<0.05$) and current ($p<0.001$) and previous smoking ($p<0.05$). VBD was strongly inversely associated with age ($p<0.001$), BMI ($p<0.001$), and positively associated with family history of breast cancer ($p = 0.023$) and older age at menarche, of 15 years or more ($p = 0.011$).

The results of linear regression analyses showed a statistically significant inverse association between creatinine-adjusted cadmium concentration in urine and VBD (β-coef: -0.077, 95% CI: -0.142, 0.013) but not with fibroglandular tissue volume (Table 2). This effect was seen in model 1: adjusted for age at mammography; BMI; family breast cancer; mammographic device; season of the year at mammography; and age at menarche. However, the effect size was relatively small, i.e. the doubled cadmium concentration was associated with roughly 5% reduction in VBD. When smoking was introduced into the multivariable analysis, this relationship was no longer significant. The VBD was found to be significantly lower in the third cadmium quartile, as compared to the first quartile in the adjusted model VBD = 6.3, 95% CI:5.8, 6.9 vs. 7.1, 95%CI:6.4, 7.8. The results were similar when the analysis was run with creatinine as an additional covariate (S1 Table).

Among the potential effect modifiers, only parity showed a statistically significant interaction with p-values for heterogeneity <0.05 found for both the outcomes (S2 Table). A significant inverse association between cadmium and VBD was found in both the adjusted models among ever pregnant women (β-coef = -0.087, 95%CI:-0.160,-0.013). The estimated mean VBD was higher among women with the lowest cadmium concentration (Q1) (7.9%, (6.2, 10.2)) than in the group with the highest levels (Q4) 7.2% (5.6, 9.1).

**Table 2. Association between urinary creatinine-adjusted cadmium concentration and percent volumetric mammographic density and fibroglandular tissue volume.**

| Outcome | | Estimate (95% Confidence interval), regression of log outcome on log Cr-adjusted cadmium concentration | | |
| --- | --- | --- | --- | --- |
| | | Unadjusted | Adjusted[1] | Adjusted[2] |
| Percent volumetric breast density | Coef. β[3] | -0.045 (-0.119,0.030) | **-0.077 (-0.142,-0.013)** | -0.059 (-0.125,0.008) |
| | exp(β)[4] | 0.956 (0.887,1.030) | **0.926 (0.868,0.987)** | 0.943 (0.882,1.008) |
| | Ratio of VBD per doubling of Cr-adj. Cd[5] | 0.969 (0.921,1.021) | **0.948 (0.907,0.991)** | 0.960 (0.917,1.006) |
| Fibroglandular tissue volume | Coef. β[3] | -0.057 (-0.123,0.009) | -0.024 (-0.088,0.040) | -0.012 (-0.078,0.055) |
| | exp(β)[4] | 0.945 (0.884,1.010) | 0.976 (0.916,1.041) | 0.988 (0.925,1.056) |
| | ratio of FG per doubling of Cr-adj. Cd[5] | 0.961 (0.918,1.007) | 0.984 (0.941,1.028) | 0.992 (0.947,1.039) |
| | | Mean (95%Confidence interval)percent volumetric mammographic density | | |
| Cadmium quartile | | | | |
| Q1:[0.008,0.38] | | 6.9 (6.3,7.6) | 7.1 (6.4,7.8) | 6.9 (6.3,7.7) |
| Q2:(0.38,0.57] | | 7.3 (6.7,7.9) | 7.6 (6.9,8.3) | 7.4 (6.8,8.2) |
| Q3:(0.57,0.79] | | 6.1 (5.6,6.7) | 6.3 (5.8,6.9) | 6.3 (5.7,6.9) |
| Q4:(0.79,3.4] | | 6.8 (6.2,7.4) | 6.5 (5.9,7.2) | 6.6 (6.0,7.2) |
| | | Mean (95%Confidence interval) fibroglandular tissue volume (cm$^3$) | | |
| Cadmium quartile | | | | |
| Q1:[0.008,0.38] | | 54.2 (50.1,58.6) | 53.1 (48.3,58.5) | 52.5 (47.6,58.0) |
| Q2:(0.38,0.57] | | 60.9 (56.3,65.8) | 59.1 (53.9,64.8) | 58.5 (53.2,64.4) |
| Q3:(0.57,0.79] | | 51.7 (47.8,55.9) | 52.4 (47.8,57.4) | 52.1 (47.5,57.2) |
| Q4:(0.79,3.4] | | 51.3 (47.5,55.5) | 52.6 (47.9,57.7) | 52.8 (48.1,58.0) |

[1] Adjusted for age at mammography; BMI; family breast cancer; mammographic device; season of the year of mammography; and age at menarche

[2] Adjusted for age at mammography; BMI; family breast cancer; mammographic device; season of the year of mammography; age at menarche and smoking

[3] Beta (β) coefficient for regression of log outcome on log Cr-adjusted Cd

[4] exp(β) for regression of outcome on Cr-adjusted Cd, which corresponds to the ratio of geometric mean outcome associated with a unit increased in log Cr-adjusted Cd

[5] exp(log(2)*β)

In a small subset of nulliparous women (n = 32), no significant association was found for VBD, but the results suggested positive associations in both crude and adjusted analyses for fibroglandular tissue volume (S2 Table). Among the nulliparous women, we observed a significant positive association between cadmium and fibroglandular tissue volume, with β-coef = 0.419, 95%CI:0.122, 0.716), and the estimated means of the fibroglandular tissue in cadmium quartiles Q1 vs Q4 of 40.9 cm$^3$(25.6, 65.3) and 72.6cm$^3$ (47.9, 109.9), respectively. The related effect size was substantial, i.e. the doubling of the cadmium concentration was associated with a 1.34-fold change in the fibroglandular tissue volume. The results of analyses stratified by smoking, menopausal status, family history of breast cancer and age groups, are presented in the supplemental tables (S3, S4, S5 and S6 Tables). No significant modifications were observed.

In order to compare the results of the present study with the findings from one of the previous studies (Adams, 2011) that showed a significant relationship between cadmium and breast density, we ran another analysis in a small subgroup of women below 45 years of age. No statistically significant associations were found, but the regression coefficients were positive both for VBD and fibroglandular tissue volume (adjusted β-coef.:0.213, p = 0.208, and 0.350, p = 0.212, respectively).

## Discussion

In the present study of middle-aged women undergoing screening mammography, we examined association between cadmium concentration in urine and volumetric mammographic density or fibroglandular tissue volume. The results have not confirmed the study hypothesis in the total study group. However, we recorded a statistically significant association between cadmium concentration in urine and the fibroglandular tissue volume in a small group of women who reported never being pregnant.

Only two previous epidemiological studies investigated the links between cadmium and mammographic breast density, and in only one of them has a positive association been found. The most recent investigation did not show any association between cadmium and mammographic density [19]. This study included women at 40–65 years of age, who were both pre- and postmenopausal, and who had breast density assessed based on routine mammographic reports, using BI-RADS classification that may have introduced some misclassification bias. Unfortunately, the report has not presented data for women younger than 45 years or for the nulliparous women. The previous study of 190 premenopausal women in US, aged 40–45, showed that each twofold increase in urine Cd concentration was associated with a statistically significant increase (1.6%) in mammographic density [18]. The effect was particularly strong among nulliparous women. Our findings for nulliparous women are consistent with Adams' observations, but our sample size was very limited, whilst our overall results and the results for parous women showed an inverse association with percent VMD. It is worth noting that in previous investigations, the participants were younger than those in our study.

In our study, we generally observed an inverse association between urinary cadmium and VBD driven by the majority subset of women who were parous. This effect was observed in the adjusted model, which included the smoking status among other important confounders. The explanation for this finding remains unknown. Residual confounding or other underlying characteristics of the subpopulations studied, which were not controlled for, may have accounted for this outcome. Another variable that strongly positively correlated with cadmium is age, but the age itself is strongly inversely associated with VBD. Therefore, in order to detect Cd and VBD association, it is critical to control for age, either through a very restricted age-range in the study design, or by adjusting appropriately. We investigated several non-linear parametrizations to adjust for age but these did not alter the association observed.

Cadmium has been identified as a potent metalloestrogen, which is thought to be a potential risk factor for breast cancer. There are several other mechanisms that make the association between cadmium exposure and MD plausible. The results of the majority of experimental studies indicate that cadmium ions may activate estrogen receptors thus mimicking estradiol activity. It has been demonstrated that cadmium initiates cell division and increases the expression of estrogen-regulated genes, such as the progesterone receptor gene. Consequently, breast cell proliferation may occur, resulting in increased MD [25–27]. Moreover, Cd interacts with antioxidant defense mechanisms through decreasing antioxidant enzyme activity, and it generates reactive oxygen species (ROS)which leads to lipid peroxidation and DNA damage. Cell damage, as a result of increased oxidative stress through chronic exposure to Cd, plays an important role in carcinogenesis and may also induce MD changes [28, 29]. Experimental studies have also shown that Cd inhibits the secretion of the connective tissue proteins, such as proteoglycan and procollagen, through fibroblasts, which potentially leads to alterations in breast architecture [30]. To sum up, by activating different biological pathways, cadmium may modify breast composition by affecting both the epithelial and stromal tissues.

To our knowledge, this study is only the third one investigating the possible association between cadmium and mammographic density. Two previous analyses used either area-based

method or BI-RADS classification for breast density assessment. Both of them are prone to subjectivity of the readers, which may introduce the misclassification bias. The strength of our study lies in the fact that it used a fully automatic and objective method for the assessment of volumetric mammographic density. The method takes into account breast thickness, and is expected to better reflect the amount of the fibroglandular tissue in the breast than the planar methods. In the present study, cadmium was measured using ICP-MS. The advantages of this method are low detection limits, wide dynamic range, high selectivity and excellent sensitivity [31]. The levels of cadmium concentration in urine that we observed were comparable to those previously reported for Polish women [2]. Moreover, the dates of urine sample collection and mammography were close in time, with the median of 24 days. Women taking HRT were not eligible for the study to avoid a strong confounding effect. Furthermore, the analysis confirmed well-established inferences for age and BMI with breast density, and for age and smoking with cadmium concentration, which supports the validity of the study.

A limitation of our study is its cross-sectional design, which does not allow to rule out reverse causation. However, it seems unlikely that the mammographic density would have influence on cadmium exposure. The population under study was not randomly selected from the general population; therefore, the study group characteristics may not reflect those in the general population of women in Lodz or in all-Poland. However, the strategy that we applied still allows for analyzing associations between biomarkers within the range of cadmium concentrations observed in the study population. Another limitation is the small number of subjects in the younger age group, hence the study was underpowered to elucidate the associations for women aged 45 years or less, i.e. the group that may be susceptible to the estrogenic effect of cadmium [18].

## Conclusions

Our study does not, in general, provide support for a positive association between cadmium concentration and mammographic density. The association of concern was found only in a very small group of women who were never pregnant, but needs verification in larger independent studies.

## Supporting information

**S1 Table. Association between urinary cadmium concentration and percent volumetric mammographic density and fibroglandular tissue volume.**
(DOCX)

**S2 Table. Association between creatinine-adjusted cadmium concentration in urine and percent volumetric mammographic density and fibroglandular tissue volume by pregnancy status.**
(DOCX)

**S3 Table. Association between creatinine-adjusted cadmium concentration in urine and percent volumetric mammographic density and fibroglandular tissue volume by smoking.**
(DOCX)

**S4 Table. Association between creatinine-adjusted cadmium concentration in urine and percent volumetric mammographic density and fibroglandular tissue volume by menopausal status.**
(DOCX)

**S5 Table. Association between creatinine-adjusted cadmium concentration in urine and percent volumetric mammographic density and fibroglandular tissue volume by family history of breast cancer.**
(DOCX)

**S6 Table. Association between creatinine-adjusted cadmium concentration in urine and percent volumetric mammographic density and fibroglandular tissue volume by age groups.**
(DOCX)

## Author Contributions

**Conceptualization:** Beata Pepłońska.

**Formal analysis:** Valerie McCormack, Paweł Kałużny.

**Funding acquisition:** Beata Pepłońska.

**Investigation:** Beata Janasik.

**Writing – original draft:** Beata Pepłońska.

**Writing – review & editing:** Beata Janasik, Valerie McCormack, Agnieszka Bukowska-Damska.

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
