## [Decision Letter · Decision Letter 0]

19 Mar 2020

PONE-D-20-03975

Cadmium and volumetric mammographic density: a cross-sectional study in Polish women

PLOS ONE

Dear Dr. Peplonska,

Thank you for submitting your manuscript to PLOS ONE. After careful consideration, we feel that it has merit but does not fully meet PLOS ONE’s publication criteria as it currently stands. Therefore, we invite you to submit a revised version of the manuscript that addresses the points raised during the review process.

Please address all reviewers' comments in a point by point response. In particular, we will be looking for a statistical response to issues of age and smoking confounding. 

We would appreciate receiving your revised manuscript by May 03 2020 11:59PM. To enhance the reproducibility of your results, we recommend that if applicable you deposit your laboratory protocols in protocols.io, where a protocol can be assigned its own identifier (DOI) such that it can be cited independently in the future. For instructions see: http://journals.plos.org/plosone/s/submission-guidelines#loc-laboratory-protocols

We look forward to receiving your revised manuscript.

Kind regards,

Jaymie Meliker, Ph.D.

Academic Editor

PLOS ONE

Journal Requirements:

2. In your Methods section, please provide additional information about the participant recruitment method and the demographic details of your participants. Please ensure you have provided sufficient details to replicate the analyses such as: a) a description of any inclusion/exclusion criteria that were applied to participant recruitment, b) a description of how participants were recruited, and c) descriptions of where participants were recruited and where the research took place.

3. Please note that PLOS does not permit references to “data not shown.” Authors should provide the relevant data within the manuscript, the Supporting Information files, or in a public repository. If the data are not a core part of the research study being presented, we ask that authors remove any references to these data.

4. Please provide a sample size and power calculation in the Methods, or discuss the reasons for not performing one before study initiation.

6. Your ethics statement must appear in the Methods section of your manuscript. If your ethics statement is written in any section besides the Methods, please move it to the Methods section and delete it from any other section. Please also ensure that your ethics statement is included in your manuscript, as the ethics section of your online submission will not be published alongside your manuscript.

Reviewers' comments:

Reviewer's Responses to Questions

**Comments to the Author**

1. Is the manuscript technically sound, and do the data support the conclusions?

Reviewer #1: Partly

Reviewer #2: Yes

2. Has the statistical analysis been performed appropriately and rigorously? 

Reviewer #1: Yes

Reviewer #2: I Don't Know

3. Have the authors made all data underlying the findings in their manuscript fully available?

Reviewer #1: Yes

Reviewer #2: Yes

4. Is the manuscript presented in an intelligible fashion and written in standard English?

Reviewer #1: Yes

Reviewer #2: Yes

5. Review Comments to the Author

Reviewer #1: This is a nice manuscript describing the cross-sectional association between urine cadmium and breast density in mammography screenings.

Here are a few comments to help the authors improve the manuscript.

1. The manuscript needs to be looked at closely for its writing. Overall, the writing is good but there are several places where the writing needs work (e.g., lines 92-97).

2. The introduction could use a little more review of previous epi studies on Cd and breast cancer incidence. Especially the prospective cohort studies which likely are closely to the true answer on the association.

3. Because creatinine is also associated with other things, some suggest running a sensitivity analysis in which creatinine is included as a covariate in the statistical model, and not dividing by it within the U-Cd measure. I would recommend doing this and reporting the results in supplemental results.

4. The nulliparous group is too small to infer anything. Those stratified results should be moved to supplemental results.

5. Figure 1 is not necessary since results are the same as table 2.

6. Most critical is the treatment of age. The authors have clearly thought a lot about age, because it is such a strong negative confounder here (positively associated with U-Cd and negatively associated with breast density). But I think this work would be markedly improved if they could identify subsets of the population in which age is not associated with breast density. Perhaps age stratifying 50-55, 56-60, etc, would still give samples of at least one hundred and would no longer see associations between age and breast density. I still would adjust for age within these groups, but the tighter strata should mitigate the age effect. They need to work to eliminate the age confounder and this is a strategy that I would likely attempt.

Reviewer #2: This manuscript, “Cadmium and volumetric mammographic density: a cross-sectional study in Polish

Women” evaluates urinary cadmium levels associated with breast density. Overall this is a well written manuscript.

The information about cadmium and breast density was well described.

The one area that needs to be addressed is the pathophysiology between cadmium and breast density. Strong factors associated with breast density are inversely associated with cadmium exposure such as age and smoking. These analyses do not adequately address this issue of competing variables. Additional statistical feedback is needed for this paper. The only thing I could think that could address this would be stratification but I don’t know if you have sufficient numbers. Although parity has not been associated with cadmium levels, it is associated with parity. And the stratification (table 3) of parity, nulliparous took this into consideration.

If the cd was evaluated using some other way to categorize such as 10th, 50th, and 90th percentiles of the creatinine-corrected urinary cadmium distribution in the overall study sample as used by Menke, would that alter the findings? See Menke A, Muntner P, Silbergeld EK, Platz EA, Guallar E. Cadmium levels in urine and mortality among U.S. adults. Environ Health Perspect. 2009;117(2):190–196. doi:10.1289/ehp.11236 for another way to categorize the data. There should be a result with cadmium as a continuous variable since categorization is an artificial way to characterize the exposure.

The other issue that is puzzling, a seems to be a proxy for something else was the seasonality of the mammography. It makes no sense that this would be significantly different based on the 2 main variation—urinary cadmium should not vary significantly over time, as this captures historical exposure; similarly breast density, as far as I know, does not vary by season. Hopefully not, but could this reflect a bias in the reading of the images?

6. PLOS authors have the option to publish the peer review history of their article (what does this mean?). If published, this will include your full peer review and any attached files.

Reviewer #1: No

Reviewer #2: No

---

## [Author Response · Author response to Decision Letter 0]

2 May 2020

Journal Requirements:

 The files have been formatted according to PLOS ONE requirements.

2. In your Methods section, please provide additional information about the participant recruitment method and the demographic details of your participants. Please ensure you have provided sufficient details to replicate the analyses such as: a) a description of any inclusion/exclusion criteria that were applied to participant recruitment, b) a description of how participants were recruited, and c) descriptions of where participants were recruited and where the research took place.

The requested information has been provided.

3. Please note that PLOS does not permit references to “data not shown.” Authors should provide the relevant data within the manuscript, the Supporting Information files, or in a public repository. If the data are not a core part of the research study being presented, we ask that authors remove any references to these data.

 The phrase in question has been removed accordingly.

4. Please provide a sample size and power calculation in the Methods, or discuss the reasons for not performing one before study initiation.

 The information about sample size and power calculation has been provided. 

The phrase has been removed. 

6. Your ethics statement must appear in the Methods section of your manuscript. If your ethics statement is written in any section besides the Methods, please move it to the Methods section and delete it from any other section. Please also ensure that your ethics statement is included in your manuscript, as the ethics section of your online submission will not be published alongside your manuscript.

The ethics statement has been moved to the Methods section.

Captions for supporting files have been included according to the PLOS ONE guidelines.

Reviewer #1: This is a nice manuscript describing the cross-sectional association between urine cadmium and breast density in mammography screenings.

Here are a few comments to help the authors improve the manuscript.

We would like to thank the Reviewer for a positive overall assessment of our manuscript. We truly appreciate the time and effort in reviewing our manuscript and the valuable comments and suggestions that will contribute to raising the scientific and editorial standard of the manuscript

1. The manuscript needs to be looked at closely for its writing. Overall, the writing is good but there are several places where the writing needs work (e.g., lines 92-97).

Respective corrections have been made and the manuscript has also been language-checked as suggested. 

2. The introduction could use a little more review of previous epi studies on Cd and breast cancer incidence. Especially the prospective cohort studies which likely are closely to the true answer on the association.

 We appreciate pointing out this issue. The information on the results of the prospective cohort studies has been included in the Introduction . 

3. Because creatinine is also associated with other things, some suggest running a sensitivity analysis in which creatinine is included as a covariate in the statistical model, and not dividing by it within the U-Cd measure. I would recommend doing this and reporting the results in supplemental results.

 The sensitivity analysis with creatinine as a confounder has been performed. The results have not changed markedly. They have been presented in the supplemental table S1. 

4. The nulliparous group is too small to infer anything. Those stratified results should be moved to supplemental results.

The table with results of the stratified analysis by parity has been moved to supplemental information (table S2). 

5. Figure 1 is not necessary since results are the same as table 2.

 Figure 1 has been removed accordingly.

6. Most critical is the treatment of age. The authors have clearly thought a lot about age, because it is such a strong negative confounder here (positively associated with U-Cd and negatively associated with breast density). But I think this work would be markedly improved if they could identify subsets of the population in which age is not associated with breast density. Perhaps age stratifying 50-55, 56-60, etc, would still give samples of at least one hundred and would no longer see associations between age and breast density. I still would adjust for age within these groups, but the tighter strata should mitigate the age effect. They need to work to eliminate the age confounder and this is a strategy that I would likely attempt.

 Following the Reviewer’s suggestion, a stratified analysis has been performed. Unfortunately, the group below age 45 was small, and therefore, we created three age group:s ≤50; >50-≤55, >55. However, the analysis has not revealed any significant findings. The results are presented in table S6 . 

Reviewer #2: This manuscript, “Cadmium and volumetric mammographic density: a cross-sectional study in Polish Women” evaluates urinary cadmium levels associated with breast density. Overall this is a well written manuscript. The information about cadmium and breast density was well described.

We would like to thank the Reviewer for his/her careful review of the manuscript and for the detailed comments and suggestions. They will certainly be helpful in raising the standard of the manuscript.

The one area that needs to be addressed is the pathophysiology between cadmium and breast density. Strong factors associated with breast density are inversely associated with cadmium exposure such as age and smoking. These analyses do not adequately address this issue of competing variables. Additional statistical feedback is needed for this paper. The only thing I could think that could address this would be stratification but I don’t know if you have sufficient numbers. Although parity has not been associated with cadmium levels, it is associated with parity. And the stratification (table 3) of parity, nulliparous took this into consideration.

To address the issue raised above, we ran additional analyses by age groups and smoking. Age and smoking were mutually adjusted besides other significant covariates. However, none of these brought forth any new findings - we did not record any significant modification of the effect. The results of these analyses are included in the supplementary tables S3 and S6. 

If the cd was evaluated using some other way to categorize such as 10th, 50th, and 90th percentiles of the creatinine-corrected urinary cadmium distribution in the overall study sample as used by Menke, would that alter the findings? See Menke A, Muntner P, Silbergeld EK, Platz EA, Guallar E. Cadmium levels in urine and mortality among U.S. adults. Environ Health Perspect. 2009;117(2):190–196. doi:10.1289/ehp.11236 for another way to categorize the data. There should be a result with cadmium as a continuous variable since categorization is an artificial way to characterize the exposure.

We thank the Reviewer for this recommendation. We ran a similar analysisas suggested; however, the results have not changed substantially. Specifically, the generalized linear model with adaptive cubic regression spline of Cadmium/Cre (similar to the one presented by Menke at al., but with a free knot location) was fit to VMD and fibroglandular volume, and produced the following graphical summary, with the solid line representing the best fit of Cd to VMD (the shaded region represents standard error of the estimate):

The above findings indicated a weak tendency towards a smaller VMD, around Cd/Cre=0.5, and little nonlinearity in cadmium effect on VMD, but it did not generally change our conclusions.

The plot for FG volume was analogous.

In the manuscript, we present data for cadmium and mammographic density measured as continuous variables. In addition, we present data in quartiles for the clarity of presentation. 

The other issue that is puzzling, a seems to be a proxy for something else was the seasonality of the mammography. It makes no sense that this would be significantly different based on the 2 main variation—urinary cadmium should not vary significantly over time, as this captures historical exposure; similarly breast density, as far as I know, does not vary by season. Hopefully not, but could this reflect a bias in the reading of the images?

We considered the season of the year as a potential confounder based on other authors’ findings. Seasonal variations in mammographic density have been suggested for premenopausal women in the study by Brisson et al. (1). We also observed that the season of the year was a significant covariate in our previous study (2). While the nature of this observation remains unknown, the season of the year in the current dataset turned out to be a significant covariate as well and was retained in the multivariate model, with Akaike criterion applied for confounders selection. 

Reference List

1. Brisson J, Berube S, Diorio C, Sinotte M, Pollak M, Masse B. Synchronized seasonal variations of mammographic breast density and plasma 25-hydroxyvitamin d. Cancer Epidemiol Biomarkers Prev. 2007;16(5):929-33.

2. Peplonska B, Bukowska A, Sobala W, Reszka E, Gromadzinska J, Wasowicz W, et al. Rotating night shift work and mammographic density. Cancer Epidemiol Biomarkers Prev. 2012;21(7):1028-37.

---

## [Editor Report · Decision Letter 1]

5 May 2020

Cadmium and volumetric mammographic density: a cross-sectional study in Polish women

PONE-D-20-03975R1

Dear Dr. Peplonska,

We are pleased to inform you that your manuscript has been judged scientifically suitable for publication and will be formally accepted for publication once it complies with all outstanding technical requirements.

With kind regards,

Jaymie Meliker, Ph.D.

Academic Editor

PLOS ONE
---

## [Editor Report · Acceptance letter]

11 May 2020

PONE-D-20-03975R1 

Cadmium and volumetric mammographic density: a cross-sectional study in Polish women 

Dear Dr. Peplonska:

I am pleased to inform you that your manuscript has been deemed suitable for publication in PLOS ONE. Congratulations! Your manuscript is now with our production department. 

With kind regards,

on behalf of

Dr. Jaymie Meliker 

Academic Editor

PLOS ONE